# PAX6 mutation alters circadian rhythm and β cell function in mice without affecting glucose tolerance

Nirav Florian Chhabra[1,2], Oana Veronica Amarie[1], Moya Wu[1,2], Anna-Lena Amend[1,2], Marina Rubey[1,2], Daniel Gradinger[1,2], Martin Irmler[1], Johannes Beckers[1,2,3], Birgit Rathkolb [1,2,4], Eckhard Wolf [2,4], Annette Feuchtinger[5], Peter Huypens [1,2], Raffaele Teperino[1,2], Jan Rozman [1,2], Gerhard K. H. Przemeck [1,2] & Martin Hrabě de Angelis [1,2,3 ✉]

The transcription factor PAX6 is involved in the development of the eye and pancreatic islets, besides being associated with sleep–wake cycles. Here, we investigated a point mutation in the RED subdomain of PAX6, previously described in a human patient, to present a comprehensive study of a homozygous *Pax6* mutation in the context of adult mammalian metabolism and circadian rhythm. Pax6[Leca2] mice lack appropriate retinal structures for light perception and do not display normal daily rhythmic changes in energy metabolism. Despite β cell dysfunction and decreased insulin secretion, mutant mice have normal glucose tolerance. This is associated with reduced hepatic glucose production possibly due to altered circadian variation in expression of clock and metabolic genes, thereby evading hyperglycemia. Hence, our findings show that while the RED subdomain is important for β cell functional maturity, the Leca2 mutation impacts peripheral metabolism via loss of circadian rhythm, thus revealing pleiotropic effects of PAX6.

[1] Helmholtz Zentrum München, Institute of Experimental Genetics and German Mouse Clinic, Neuherberg, Germany. [2] German Center for Diabetes Research (DZD), Neuherberg, Germany. [3] Chair of Experimental Genetics, Center of Life and Food Sciences, Weihenstephan, Technische Universität München, Freising, Germany. [4] Ludwig-Maximilians-Universität München, Gene Center, Chair for Molecular Animal Breeding and Biotechnology, Munich, Germany. [5] Helmholtz Zentrum München, Research Unit Analytical Pathology, Neuherberg, Germany. ✉email: hrabe@helmholtz-muenchen.de

Paired box domain 6 (PAX6) is a transcription factor that is aptly termed as a pleiotropic factor for its importance in forebrain pattering, development of the eye, the olfactory system and of pancreatic islets[1–4]. The protein consists of two DNA-binding domains: an N-terminal paired domain and a C-terminal homeodomain[5], together with a transactivation domain rich of proline, serine, and threonine (PST). The N-terminal paired domain is further divided into two different helix-turn-helix subdomains (referred to as PAI and RED). The *Pax6* gene is highly conserved across species, with mouse and human sharing an identical amino acid sequence within the N-terminal paired domain and C-terminal homeodomain[6]. The murine heterozygous small eye mutant Pax6[Sey] that lacks the C-terminal homeodomain is characterized by small eyes and iris hypoplasia[3], which is mirrored in human patients afflicted with aniridia due to heterozygous *PAX6* mutation[7]. Homozygous Pax6[Sey] mice die at birth and their craniofacial abnormalities are coupled with a decrease in the number of all major islet cell types[1,3], suggesting abnormality in pancreas development. The Pax6[Aey18] mouse model (lacking the PAI subdomain) displays severe pancreatic α cell depletion as compared to a modest decrease observed in the Pax6[Sey-Neu] model (lacking the PST domain)[8]. Correspondingly in humans, mutations in *PAX6* can result in varied abnormalities of the eye such as Peter's anomaly and aniridia, the metabolic disease diabetes or a combination of both, depending on the respective mutation[7,9–11]. Hence, it is warranted to elucidate phenotypic differences pertaining to the location of mutations in the *Pax6* gene, thereby comparatively describing functional aspects of the independent DNA-binding domains[8,12,13].

In the context of metabolism, studies have reported the use of in vivo loss-of-function mouse models and confirmed the importance of PAX6 in the development of endocrine cells[2,14]. Furthermore, PAX6 directly regulates islet hormonal gene expression such as insulin, glucagon, and somatostatin[1] and affects insulin secretion by β cells[15], hence indicating a pertinent role of PAX6 in the adult stage. Recently, two independent studies analyzed the loss of PAX6 specifically in adult β cells and demonstrated an essential role for PAX6 in β cell identity by repressing "disallowed genes" and other islet hormonal markers[16,17].

While various tissue specific mouse models have revealed important molecular pathways involving PAX6, whole body mutants illuminate phenotypic differences with a systemic perspective. Previously, a point mutation (R128C) in the RED subdomain of the N-terminal paired domain described in a human patient[18], was studied in the mouse model Pax6[Leca2] with regards to brain development[19,20]. The authors reported homozygous viability of Pax6[Leca2] mice with a normal span of life[20]. As yet, most mice homozygous for any *Pax6* mutation either die prenatally or shortly after birth[1,2,8,21,22]. Therefore, Pax6[Leca2] mice present a unique opportunity to investigate effects of a mutated RED subdomain of PAX6 on the adult mammalian physiology.

In this study, we demonstrate that Pax6[Leca2] mice have disorganized melanopsin positive intrinsically photosensitive retinal ganglionic cells (ipRGCs) in eye-like structures and display loss of circadian rhythm. Unexpectedly, Pax6[Leca2] mice have significantly lower blood glucose levels as compared to wild-type mice and normal glucose disposal, while displaying hypoinsulinemia and altered β cell functional maturity. Further in vivo investigations revealed that these mice have reduced hepatic glucose production and altered liver function possibly due to loss of rhythmicity of metabolic processes. Since light-mediated entrainment of the central clock via the superchiasmatic nucleus (SCN) in the hypothalamus occurs as consequence of light perception via ipRGCs[23], this study hints at a possible role of PAX6 in the correct organization and formation of ipRGCs and its indirect effect on the circadian rhythm[24]. Hence, our data suggest the interaction of various biological pathways affected by the pleiotropy of PAX6, which consequently seem to protect Pax6[Leca2] mice from hyperglycemia in the absence of insulin increment.

## Results

### Structural disorganization of the retina and lack of optic nerve in Pax6[Leca2] mice.
Previously, a study reported lack of eyes in homozygous Pax6[Leca2] mutants at the embryonic stage, while in adult mice eyelids are closed and conceal any eye structures that may be present[20]. We performed MRI imaging of the eye and found small eye-like structures, suggesting accumulation of cells and tissues (Supplementary Fig. 1a). To determine the nature of the tissue and cells in the eye sockets of these mice, we isolated eye-like structures and investigated the state of ipRGCs. We carried out triple immunostaining for NEUN (also known as RBFOX3), which is expressed by neural cells, BRN3A marking retinal ganglionic cells[25], and melanopsin (Fig. 1a–f). In wild-type mice, we observed a layer of BRN3A[+]/NEUN[+] cells (Fig. 1g, arrows) and melanopsin[+] cells co-expressing BRN3A (Fig. 1g′, arrowheads) in the retinas indicating intact structures for light sensing and information relay. On the contrary, Pax6[Leca2] mice did not display normal retinal-like structures or a layer of RGCs as observed for wild-type eyes. Instead, we observed single NEUN[+] and melanopsin[+] cells (Fig. 1h, h′, arrows) as well as several cells co-expressing melanopsin/BRN3A and BRN3A/NEUN (Fig. 1h′, arrowheads), scattered in a random fashion throughout the eye structure. These data indicate lack of functional units for light perception in Pax6[Leca2] mice. Next, we investigated the presence of the retinal nerve innervations into the hypothalamus. Interestingly, the mutant mice not only lacked the optic chiasm but also the optic nerve, indicating absence of an essential component of the retinohypothalamic tract (Supplementary Fig. 1b). This strongly suggested a lack of relay of relevant information to the hypothalamus for light entrainment, which might result in disturbance of the central clock. Therefore, we extracted the hypothalamus from mice at four different circadian time points (Zeitgebers, ZTs), with an interval of 6 h starting at ZT0 (06:00) and analyzed changes in gene expression over time. Pax6[Leca2] mice failed to show the normal pattern of circadian variation in important genes associated with the core circadian machinery in contrast to wild-type samples that showed the expected temporal oscillations (Fig. 1i). Moreover, we observed high intra-group variability in Pax6[Leca2] mice, which is suggestive of loss of synchronicity and temporal coordination with light (Fig. 1i).

### Loss of circadian rhythm in Pax6[Leca2] mice.
Retinal images and lack of temporal changes in hypothalamic expressions of clock genes in Pax6[Leca2] mice strongly implied alterations in circadian rhythm. To investigate whether this translates into changes in rhythmicity of metabolic processes, we kept mice in metabolic cages to monitor locomotor activity, feeding behavior and energy expenditure over a period of 3 days (after 1 day of acclimation). Under normal conditions, wild-type mice increase their oxygen consumption (VO₂) during the dark (active) phase and decrease it during the light (inactive) phase, reflecting predominant feeding and fasting states, respectively (Fig. 2a, b). In contrast, Pax6[Leca2] mice maintain similar energy expenditure between the phases (Fig. 2a, b). Likewise, we observed a lack of significant change in locomotor activity between the light and dark phases of Pax6[Leca2] mice (Fig. 2c, d). These results were indicative of loss of circadian rhythm and modifications in feeding behavior. Indeed, although Pax6[Leca2] mice displayed some changes in food intake during the day, the oscillations were blunted and inconsistent compared to wild-type littermates (Supplementary Fig. 2a, b), and without any change in cumulative food intake (Supplementary

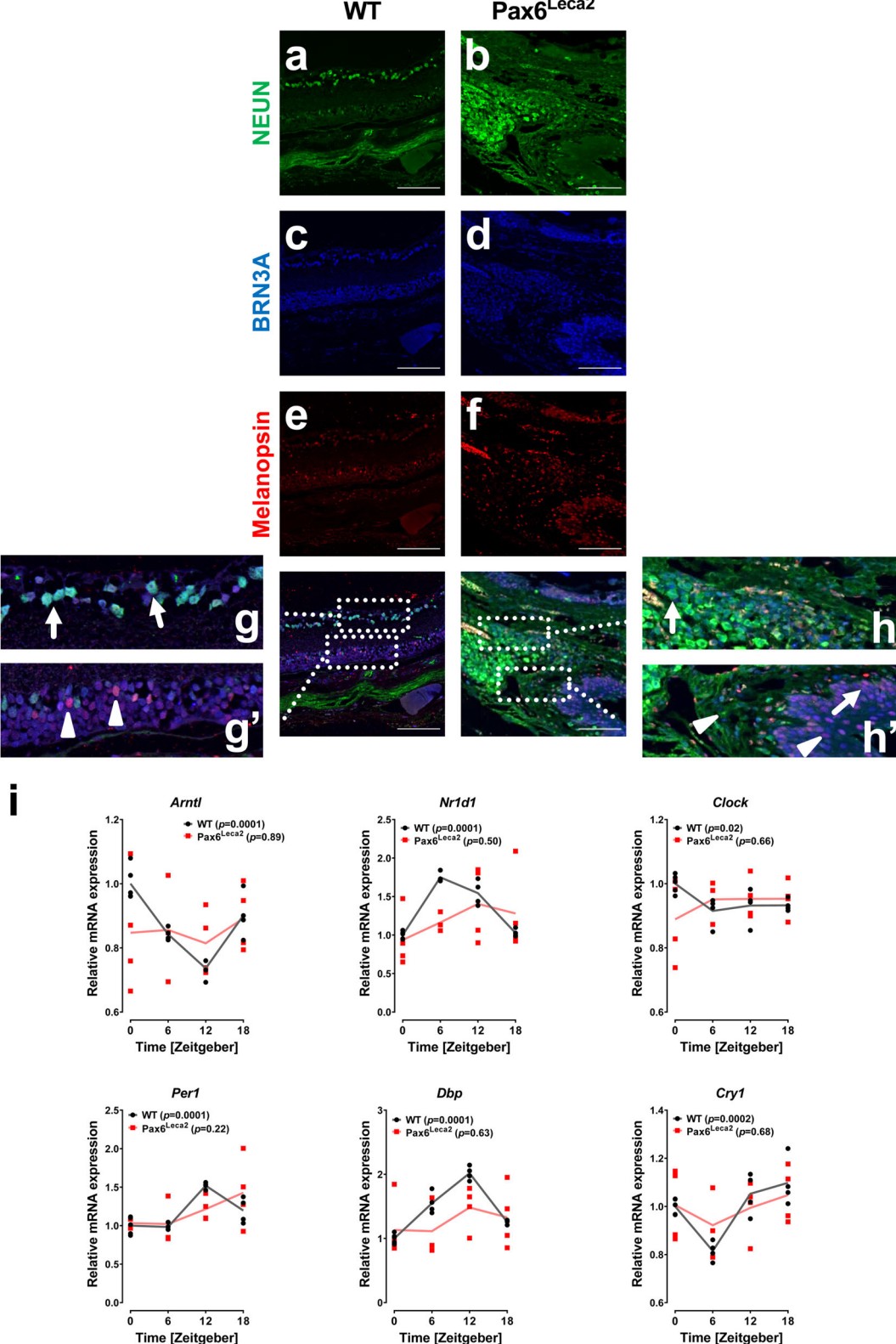

**Fig. 1 Structural disorganization of the retina in homozygous Pax6$^{Leca2}$ mice. a–h**' Representative immunofluorescence images displaying **a**, **b** expression of NEUN, **c**, **d** BRN3A, **e**, **f** Melanopsin, and **g–h**' merged images of eyes and eye-like structures of 20-week-old male mice. $n = 4$. Arrow and arrowheads depict cell types as specified in the text. Scale bar, 100 μm. **i** Relative mRNA expressions of circadian genes in hypothalamus at different ZTs (normalized to WT ZT0) as specified in 14–16-week-old male mice. $n = 4$ (WT ZT0 $n = 5$, Leca2 ZT6 $n = 3$). $p$ values in parentheses described in the graphs were acquired by applying CircWave analysis.

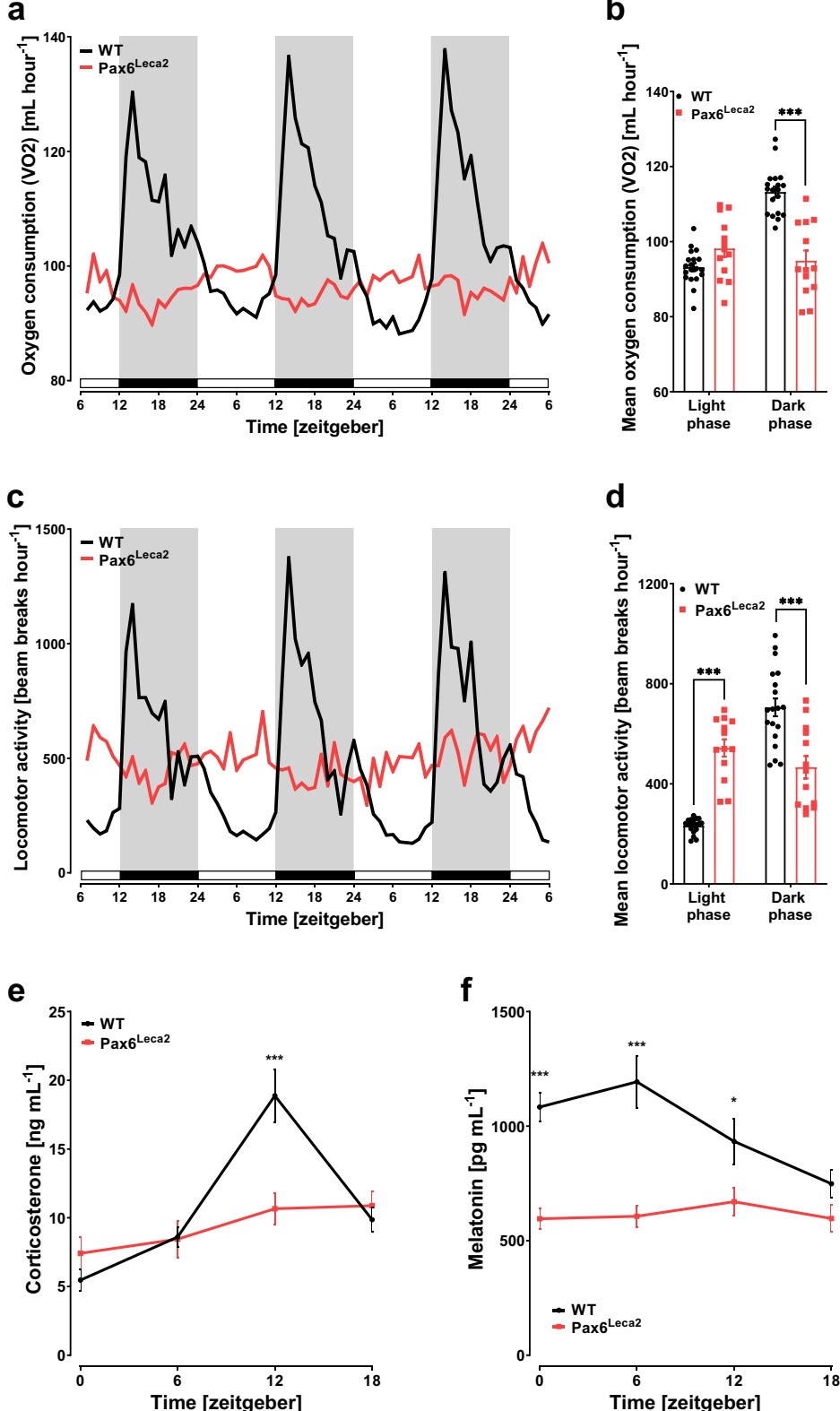

**Fig. 2 Loss of circadian rhythm in Pax6Leca2 mice. a–d** Indirect calorimetry measurements taken over 72 h displaying (WT $n = 19$, Leca2 $n = 13$) **a** oxygen consumption, **b** average oxygen consumption (***$p < 0.001$ one-way Welch's ANOVA followed by Dunnett's post-hoc test), **c** locomotor activity, and **d** average locomotor activity (***$p < 0.001$ one-way Welch's ANOVA followed by Dunnett's post-hoc test). Gray shade and black bars depict lights off and white shade and bars depict lights on. Fourteen-week-old male mice were used for this study. Temporal measurements of plasma corticosterone in **e** and melatonin in **f** at different ZTs as specified (WT $n = 8$, Leca2 $n = 7$). (*$p < 0.05$, **$p < 0.01$, ***$p < 0.001$ two-way ANOVA followed by Bonferroni's post-hoc test. Thirteen- to fifteen-week-old male mice were used for this study. Error bars display ±s.e.m.

Fig. 2c). A similar pattern was observed for oscillations in the respiratory exchange ratio (RER) (Supplementary Fig. 2d, e), which indicated changes in metabolic substrate utilization and an altered metabolic flexibility. To confirm the loss of circadian rhythm in Pax6[Leca2] mice, we collected plasma to measure parameters and assessed the state of diurnal rhythm at four different ZTs. In-line with oxygen consumption, plasma corticosterone levels in Pax6[Leca2] mice did not show any significant change throughout the day, while a clear feeding and activity anticipatory peak was visible for wild-type mice at ZT12 (6 pm) before the onset of the dark phase (Fig. 2e). Similarly, we observed low and arrhythmic circulating amounts of melatonin, the main hormonal regulator for circadian rhythm[26] in Pax6[Leca2] mice (Fig. 2f). Hypoplasia of the pineal gland has been reported for some patients with PAX6 mutations[27,28]. Thus, we carried out micro-computed tomography (µCT) and volumetric analysis of the pineal gland and found a strong trend towards decrease in pineal gland volume in mutant mice (Supplementary Fig. 2f). Taken together, these data demonstrate that Pax6[Leca2] mice display loss of circadian rhythm.

**Pax6[Leca2] mice display normal glucose tolerance in spite of impaired insulin secretion.** Given the strong association between circadian rhythm and glucose metabolism[29], we measured blood glucose levels and body weight. While heterozygous littermates showed a comparable body weight and blood glucose (Supplementary Fig. 3a, b), Pax6[Leca2] mice persistently displayed lower levels as compared to their wild-type littermates throughout the day (Fig. 3a). Furthermore, these mice showed lower body weight (Fig. 3b) suggesting changes in whole body physiology. We then conducted an oral glucose tolerance test (oGTT) and unlike other Pax6 mutant models that displayed glucose intolerance[30,31], Pax6[Leca2] mice showed normal glucose disposal (Fig. 3c) despite significantly reduced insulin levels during the glucose challenge (Fig. 3d). To ascertain whether low insulin levels observed in vivo were a consequence of reduced insulin, we investigated insulin content in Pax6[Leca2] islets. Whole islet insulin content was significantly decreased in these mice, without any change in glucagon content (Fig. 3e, f). PAX6 has been shown to regulate insulin secretion[32], therefore we utilized in vitro tools to demonstrate the capacity of insulin secretion in Pax6[Leca2] β cells. Isolated islets of Pax6[Leca2] mice when stimulated with high glucose and the insulin secretagogue exendin-4, failed to effectively increase insulin secretion as compared to wild types, thereby recapitulating results seen in vivo (Fig. 3g). Thus, Pax6[Leca2] mice have normal glucose tolerance, despite reduced amounts of insulin and impaired insulin secretion.

**Leca2 mutation leads to changes in the islet transcriptome.** To further investigate the molecular changes associated to the functional alterations, we performed mRNA expression analysis by employing microarrays on isolated islets. Over 1300 genes (Fig. 4a, Supplementary Data 1) were differentially expressed in mutant mice. To identify direct PAX6 targets in the dataset, we compared it with available ChIP-Seq data (GSE87530) from a mouse pancreatic β cell line[16] and found that ~23% of the genes downregulated in mutant islets were direct PAX6 targets as compared to about ~8% of the upregulated genes (Fig. 4b, Supplementary Data 1). In keeping with reported findings[20], Pax6 expression was significantly increased in Pax6[Leca2] islets as compared to wild-type islets (Fig. 4c), in concert with a similar increase in PAX6 protein levels (Supplementary Fig. 4a) with a normal nuclear expression within the islet (Supplementary Fig. 4b). In-line with the reduced islet insulin content, expression of both insulin gene isoforms Ins1 and Ins2 were significantly downregulated (Supplementary Data 1, Fig. 4c). Furthermore, we

found significant changes in expression of genes associated with glycolysis (G6pc2), TCA cycle (Pcx) and insulin secretion (Ffar1, Glp1r, Rapgef4, and Sytl4) in islets of Pax6[Leca2] mice (Fig. 4c, Supplementary Data 1). Notably, we detected a trend towards reduced expression of Mafa and a significant reduction in expression of Ucn3 (Fig. 4c, Supplementary Data 1) in Pax6[Leca2] islets. PAX6 binds to the enhancer region of Slc2a2[16] (encoding GLUT2), whose expression is lost in Pax6-null mice[33]. Remarkably, insulin-positive β cells displayed dramatic loss in protein expression of GLUT2 (Fig. 4d). Collectively, these findings hinted at loss of β cell functional maturity. In support of this notion, we observed increased expressions of certain "disallowed genes" such as Clec2d, Sorbs3, Arhgap29, Pld1, Pla1a, Abcg2, Lrp1[17,34], including the early endocrine marker Neurog3[35] (Supplementary Data 1) in Pax6[Leca2] islets.

PAX6 is known to maintain terminally differentiated states across several cell types[36,37] and recent studies have shown that Pax6-null β cells in adult islets may switch to alternate cell fates[38,39]. Stemming from the observed reduction in insulin content and the expression of β cell maturity markers as well as increased expression of disallowed genes in mutant islets, we investigated whether the Leca2 mutation affects islet cell population. We investigated the state of gene expression of the islet hormones and observed no significant change in the expression of Sst, Ppy, or Ghrl (Supplementary Fig. 4c). However, Pax6[Leca2] mice displayed reduced expression levels of Gcg. This prompted us to measure α and β cell mass. Although Pax6[Leca2] mice display dramatic reduction in insulin content, we observed significantly increased β cell mass and decreased α cell mass, effectively increasing β to α cell ratio in these mice (Fig. 4e). Accordingly, we found significantly increased proliferation marker Ki-67 in insulin[+] cells of mutant islets (Supplementary Fig. 4d, e). Nevertheless, in agreement with comparable islet glucagon content between both groups, we also found similar levels of circulating glucagon levels under fasted condition (Supplementary Fig. 4f). Therefore, the Leca2 mutation likely exerts negative effects on the β cell transcriptome, hence suggesting an important role of the RED subdomain in maintenance of β cell function and insulin secretion.

**Decreased hepatic glucose production in Pax6[Leca2] mice.** The ability of Pax6[Leca2] mice to maintain low blood glucose levels albeit reduced circulating insulin levels, was a surprising result. Changes in body composition, such as decrease in fat mass and/or increase in lean mass may contribute to quicker glucose disposal. Therefore, we measured body composition of these mice at 14 weeks of age and found that Pax6[Leca2] mice showed a significant ($p < 0.0001$) but mild increase in fat mass (Fig. 5a), while a significant decrease in lean mass ($p < 0.0001$) (Fig. 5b) when adjusted to body weight, thus excluding body composition as a major contributor to normal glucose tolerance in these mice. Next, we speculated that increased insulin sensitivity might have influenced quicker glucose disposal in Pax6[Leca2] mice. Indeed, an insulin tolerance test (ITT) clearly demonstrated quicker blood glucose disposal in these mice (Fig. 5c). To explore the origin of this acquired insulin sensitivity, we carried out a hyperinsulinemic-euglycemic clamp (Supplementary Fig. 5a), which revealed an increased glucose infusion rate (GIR) in Pax6[Leca2] mice (Fig. 5d). Next, we wondered whether any change in glucose uptake or turnover might have contributed to the increased GIR. Interestingly, no significant change in adipose tissue and muscle glucose uptake (Fig. 5e) as well as total glucose turnover was found between the groups (Supplementary Fig. 5b). Additionally, we did not observe any significant difference in urinary glucose loss at the end of the experiment between the groups (Supplementary Fig. 5c). Surprisingly, Pax6[Leca2] mice displayed a dramatic decrease in hepatic glucose production

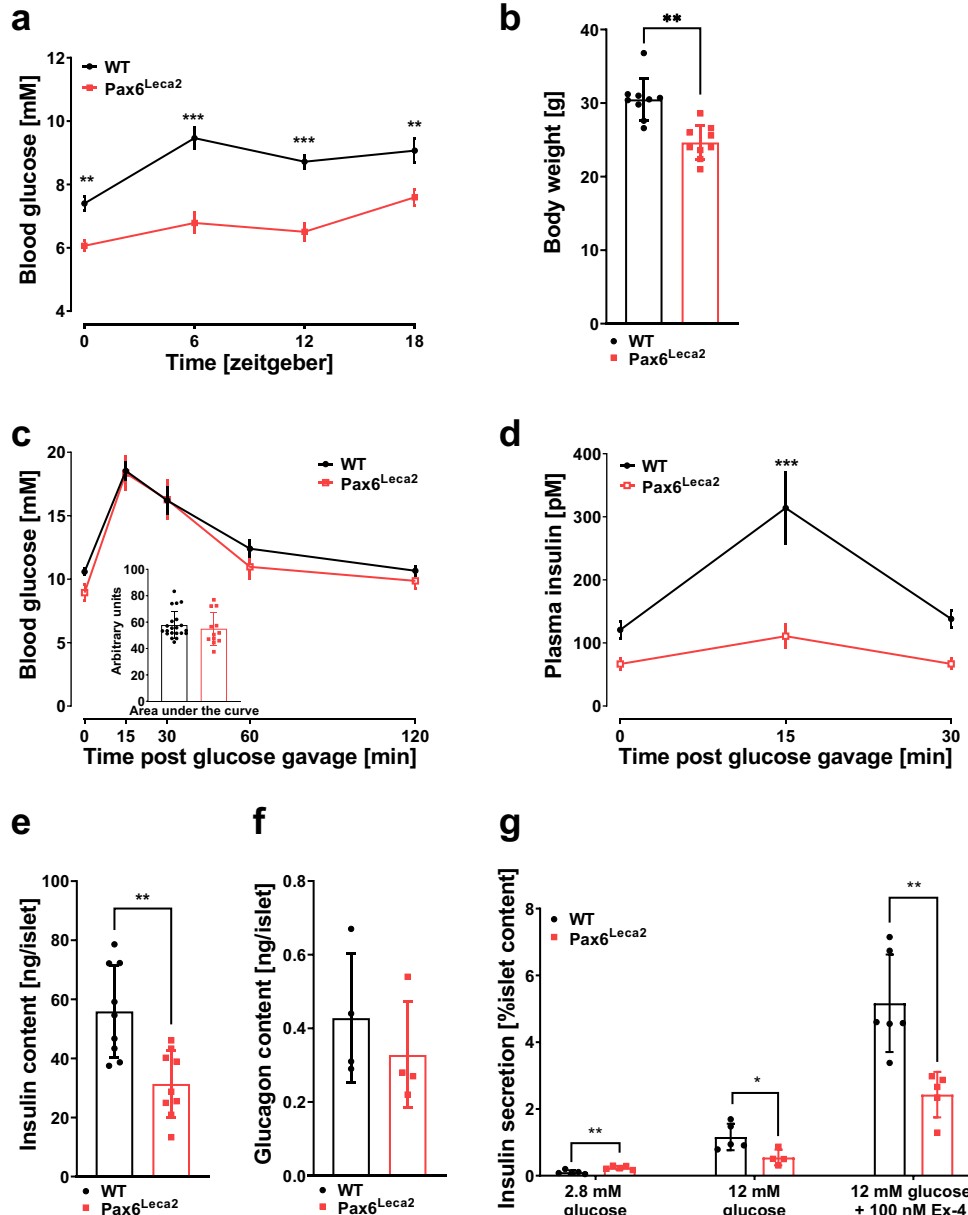

**Fig. 3 Pax6^Leca2 mice display normal glucose tolerance in spite of impaired insulin secretion. a** Temporal measurement of blood glucose levels.
**b** Average *ad libitum* body weight. Fourteen-week-old mice were used for these experiments. $n = 9$, $*p < 0.05$, $**p < 0.01$, $***p < 0.001$ Student's *t*-test and two-way ANOVA followed by Bonferroni's post-hoc test. **c** Oral glucose tolerance test (oGTT) and **d** insulin measurements during the oGTT. WT $n = 21$ (**c**), $n = 19$ (**d**), Leca2 $n = 12$, $**p < 0.01$, $***p < 0.001$ two-way ANOVA followed by Bonferroni's post-hoc test. Total islet protein content of insulin ($n = 9$) in **e** and glucagon ($n = 4$) in **f**. $*p < 0.05$, $**p < 0.01$ Student's *t*-test. **g** Glucose stimulated insulin secretion assay. WT $n = 6$, Leca2 $n = 4$, $*p < 0.05$, $**p < 0.01$ Student's *t*-test. Ten- to twelve-week-old mice were used for these experiments. Error bars display ±s.e.m. in **a**, **c**, **d**, rest ±s.d.

during the clamp experiment (Fig. 5f), indicating that reduced hepatic glucose production may be driving the increased GIR. To confirm this, we carried out western blot analysis of insulin induced AKT phosphorylation (pAKT) expression in liver samples and found that the both groups similarly increased pAKT expression upon insulin stimulation (Supplementary Fig. 5d), suggesting that the ITT may not display increased insulin sensitivity per se but rather a reflection of reduced hepatic function.

**Lack of rhythmic changes in clock and metabolic genes in the liver of Pax6^Leca2 mice.** Since the clamp experiment signified alterations in liver function, we hypothesized that this might be associated with changes in circadian rhythm. Hence, we examined

the state of clock genes and observed that Pax6^Leca2 mice displayed an asynchronous gene expression pattern in the liver (Fig. 6a). A similar pattern was observed for expression of genes associated with glycolysis (*Gck* and *Pklr*), gluconeogenesis (*Pck1* and *G6pc*) and *Slc2a2* and *Gys2* (Fig. 6b). Moreover, we carried out biochemical analysis of plasma samples acquired from the vena cava blood of mice that did not show changes in glycerol and nonesterified fatty acids (NEFAs) between mutants and wild types (Supplementary table 1), ruling out any lack of gluconeogenic substrates. Next, we wondered whether Pax6^Leca2 mice have dampened capacity for hepatic glucose production and tested it by intraperitoneal pyruvate tolerance tests. Interestingly, pyruvate administration after a 6-h fasting period did not produce any differences between the groups (Supplementary Fig. 6a). Finally,

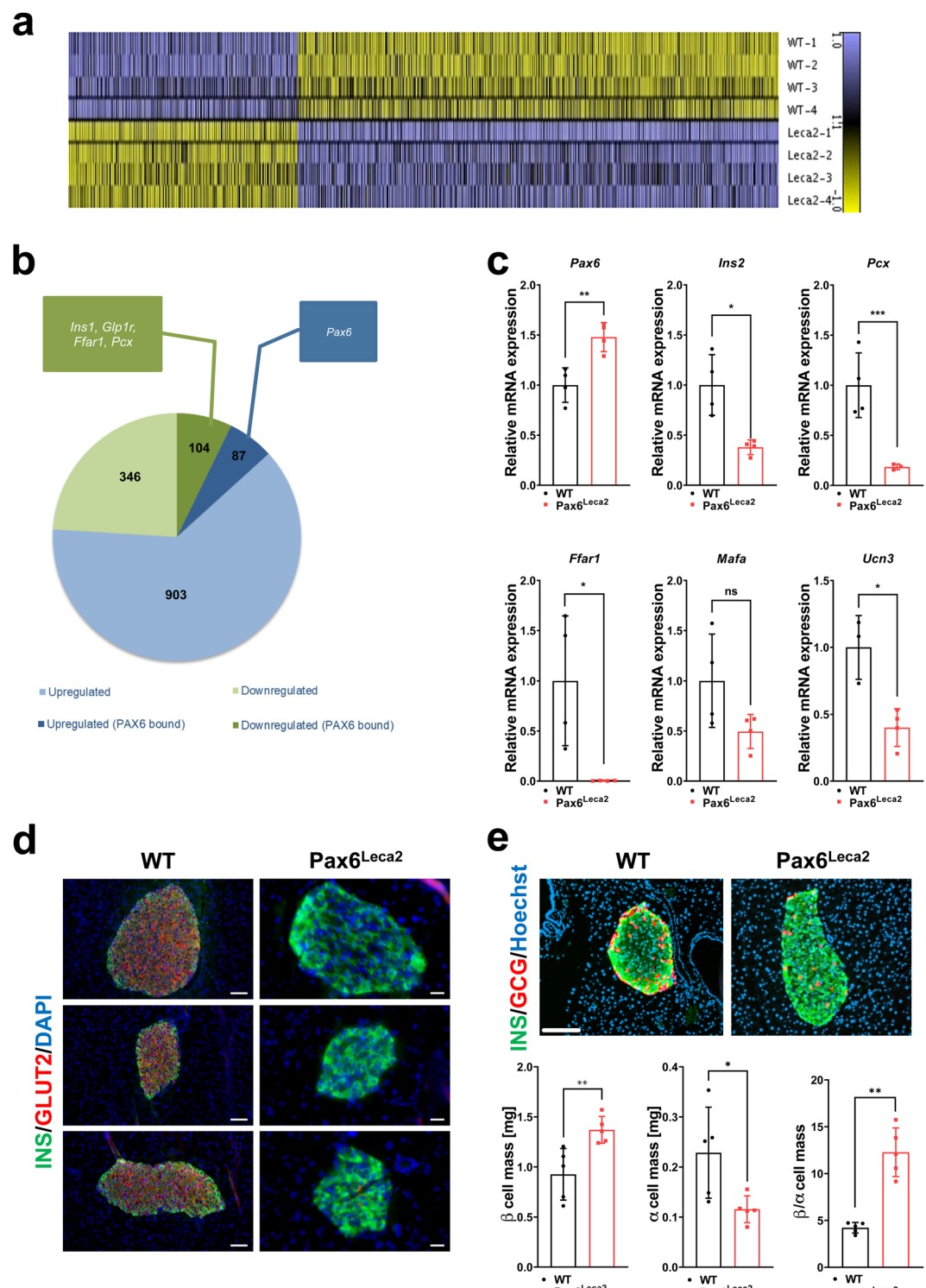

**Fig. 4 Leca2 mutation leads to changes in islet transcriptome. a** Heat map displaying differentially expressed genes (DEGs) in isolated islets. Genes were filtered for fold-change >1.5× and FDR < 10%. $n = 4$. **b** A comparative analysis of DEGs in Leca2 mutants and known PAX6 targets. **c** Relative islet mRNA expression of *Pax6*, *Ins2*, *Pcx*, *Ffar1*, *Mafa*, and *Ucn3*. $n = 4$ (WT *Ucn3* $n = 3$, Leca2 *Pcx* $n = 3$), ns non-significant, *$p < 0.05$, **$p < 0.01$, ***$p < 0.001$ Welch's and Student's *t*-test. **d** Representative immunofluorescence images for GLUT2 in pancreatic islets. $n = 3$. Scale bar, 50 μm. **e** Representative immunofluorescence images displaying insulin and glucagon-positive cells in islets and quantifications of α and β mass, and their ratio. $n = 5$, >80 islets per mouse were analyzed. Scale bar, 100 μm. *$p < 0.05$, **$p < 0.01$ Welch's *t*-test and Student's *t*-test. Ten- to twelve-week-old mice were used for all experiments. Error bars display ±s.d.

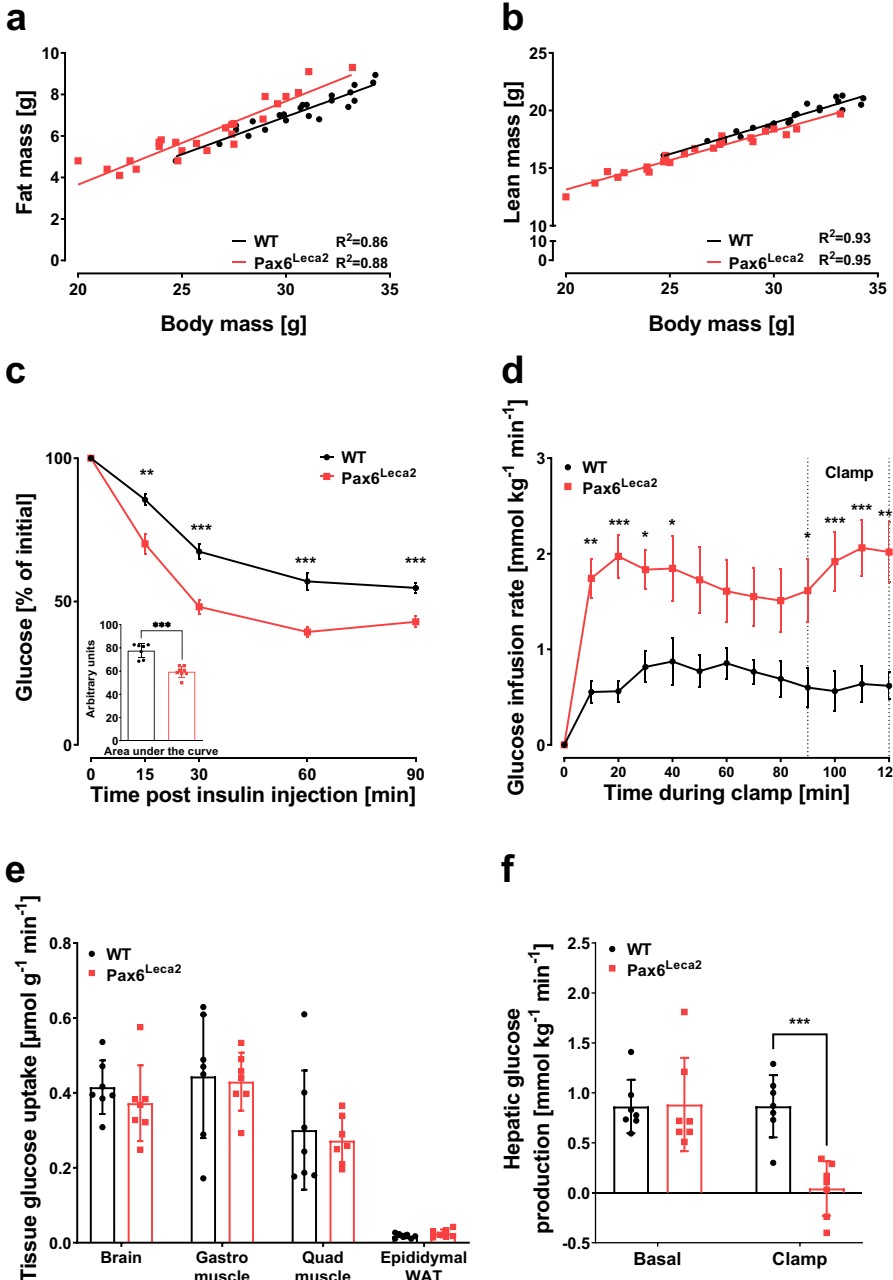

**Fig. 5 Decreased hepatic glucose production in Pax6[Leca2] mice. a**, **b** Linear regression model displaying **a** fat mass and **b** lean mass plotted against body mass of 14-week-old male mice. WT $n = 23$. Leca2 $n = 25$. **c** Intraperitoneal insulin tolerance test. WT $n = 7$, Leca2 $n = 8$, **$p < 0.01$, ***$p < 0.001$ Welch's $t$-test and two-way ANOVA followed by Bonferroni's post-hoc test. **d**–**f** Results from hyperinsulinemic-euglycemic clamp displaying **d** Glucose infusion rate (*$p < 0.05$, **$p < 0.01$, ***$p < 0.001$ two-way ANOVA followed by Bonferroni's post-hoc test), **e** glucose uptake by peripheral tissues, and **f** hepatic glucose production (***$p < 0.001$ one-way ANOVA followed by Bonferroni's post-hoc test). $n = 7$, 12–14-week-old male mice were used. Error bars display ±s.e.m. in **c**, **d** and ±s.d. in **e**–**f**.

we checked plasma samples from these mice for any elevation in ketone bodies[40]. Indeed, β-hydroxybutyrate was increased 2-fold in Pax6[Leca2] mice as compared to wild types after a 6-h fasting period (Supplementary Fig. 6b). Thus, Pax6[Leca2] mice have normal hepatic capacity for glucose output, however under physiological conditions, seem to be unable to produce enough glucose, possibly due to loss of rhythmic changes in metabolic processes.

## Discussion
Although the role of PAX6 has been well established in development of the islets[1,2,14] and recently in adult β cells[16,17], this study demonstrates multifaceted effects spanning several organ systems and highlights new aspects regarding PAX6. A single point mutation in the RED subdomain on one hand, is sufficient to cause β cell dysfunction and reduced insulin secretion due to specific changes in the islet transcriptome. On the other, the mutation leads to loss of circadian rhythm in Pax6[Leca2] mice likely due to disorganization of ipRGCs and a lack of optic nerve innervations. This in turn seems to result in disturbances in liver metabolism and altered metabolic flexibility. However, lack of circadian variations associated with liver function lead to lower blood glucose levels rather than a hyperglycemic state (Fig. 7).

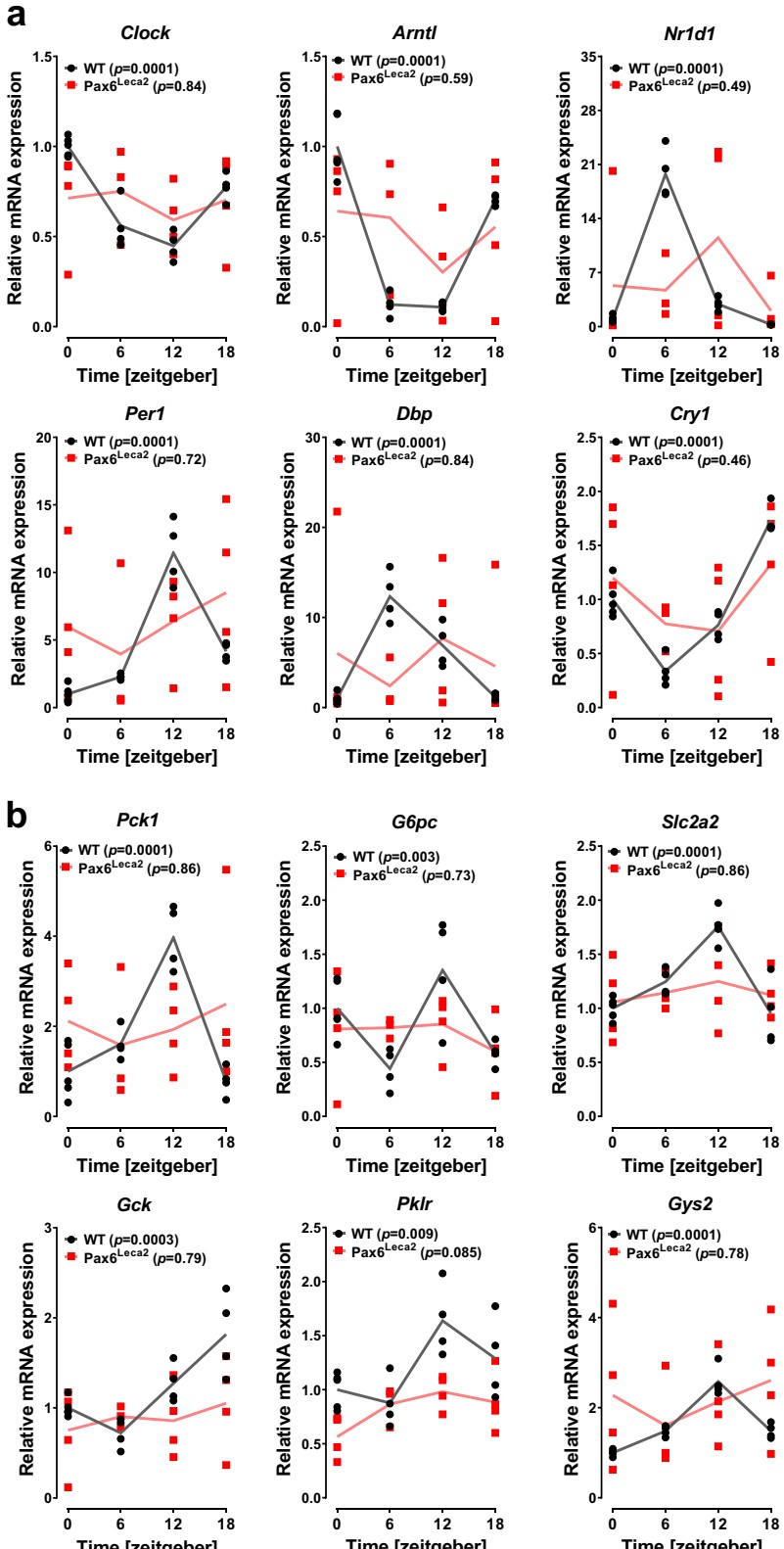

**Fig. 6 Lack of rhythmic changes in clock and metabolic genes in the liver of Pax6$^{Leca2}$ mice. a, b** Relative liver mRNA expressions of **a** circadian genes and **b** glucose metabolism at different ZTs (normalized to WT ZT0) as specified in 14–16-week-old male mice. $n = 4$ (WT ZT0 $n = 5$, Leca2 ZT6 $n = 3$). $p$ values in parentheses described in the graphs were acquired by applying CircWave analysis.

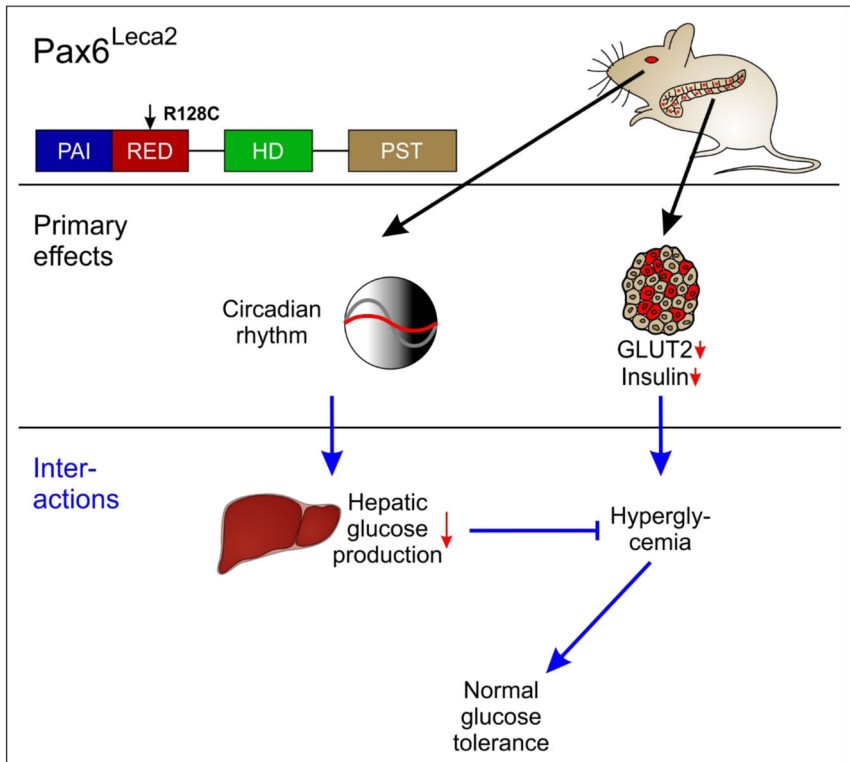

**Fig. 7 Graphical summary.** Primarily, the Leca2 mutation seems to directly affect pancreatic islets and eye development. However, decreased circulating insulin does not produce a hyperglycemic phenotype in Pax6[Leca2] mice. Instead, decreased hepatic glucose production as a consequence of loss of circadian rhythm results in normal glucose tolerance.

Most homozygous *Pax6* mouse models carrying a point mutation and studied so far for pancreatic function and morphology either die prenatally or shortly after birth[8,21,31] with the exception of the Pax6[14Neu] model, that lives a normal span of life and does not show any observable pancreatic phenotype[8,12]. On the contrary, the homozygous Pax6[Leca2] model carrying a human mutation[18] is the only viable mouse model with a mutation in the RED subdomain of PAX6. An astonishing feature of the Pax6[Leca2] model is the loss of circadian rhythm. Although much less is known about PAX6 in this regard, it may be implicated in establishing and maintaining diurnal cycle via two essential developmental pathways. First, PAX6 is a well-known initiator of eye development and eyes harbor melanopsin expressing ipRGCs that provide the primary signal for synchronizing internal clock to the environment. Importantly, the genetic background of the Pax6[Leca2] mice is C3HeB/FeJ, hence they suffer photoreceptor loss that occurs rapidly with onset at postnatal day 8 (P8) and is nearly complete by P21, due to inactivation of the rod photoreceptor cGMP phosphodiesterase six β subunit (*Pde6b*) gene[41–43]. Therefore, the visual pathway responsible for image formation is distorted in mutant and wild-type mice alike. Nonetheless, Pax6[Leca2] mice lack appropriate retinal structures, which suggests that light-mediated entrainment of circadian rhythm is likely compromised. A second pathway with which PAX6 may affect circadian rhythm is by its implication in the development of the pineal gland. Previous studies have reported hypoplasia of the gland and altered sleep-wake cycle in patients with *PAX6* mutations[27,28]. However, melatonin secretion lies downstream of the SCN[26] and melatonin like corticosterone does not display a rhythmic change in Pax6[Leca2] mice, which in part may be due to reduced pineal gland volume. Nevertheless, lack of functional units for light perception and the transfer of information on the time of day via the optic nerve seems to be

the primary reason behind the lack of a circadian rhythm. Consistently, optic nerve malformations have been reported in humans with *PAX6* mutations. Of note, several clinical variations at the R128 position of PAX6, namely R128H, R128L, R128S (NCBI database[44]), and R128P[45] have been reported, indicating a propensity for mutations at the R128 site in humans and warrant further investigation[46,47].

Pax6[Leca2] islets have reduced islet insulin content and markedly different expression of various genes associated with β cell function. Eliminating *Pax6* specifically in β cells leads to modifications in cell identity and consequently changes in islet cell type composition due to the increase in expression of genes that are normally repressed in β cells[16,17]. In contrast, Pax6[Leca2] mice do not show upregulation of genes enriched in other islet cell types. Instead, we specifically observed changes affecting β cell maturity such as decreased gene expression of *Ucn3*[48], increase in the expression of certain disallowed genes[34] and loss of GLUT2 expression in islets of Pax6[Leca2] mice. Moreover, a large number of known direct PAX6 targets in β cells were found to be regulated in mutant islets, suggesting that in some measure, the PAX6 RED subdomain confers β cell properties to adult pancreatic islets.

Increase in β cell mass in is an interesting observation that has not yet been reported for any other *Pax6* mutant model. PAX6 acts as both a repressor and an activator[39], and the Leca2 mutation has been implicated in reducing the antiproliferative property of PAX6[20]. Interestingly, previous studies have demonstrated that the Leca2 mutation reduced activity of PAX6 on the P5aCON site, which is driven by the alternative spliced isoform PAX6-5a[20,49]. A recent study showed that reduced amounts of the *Pax6-5a* isoform can abolish transcription of *Gcg* and associated PAX6 direct targets such *Mafb* and *Arx*[50]. This may in part explain the observed reduction in *Gcg* transcription.

However, several studies have revealed that mice display considerable resistance to α cell ablation[51–53], which supports lack of alterations in glucagon islet content and fasting glucagon levels in Pax6[Leca2] mice.

An astonishing feature of the Pax6[Leca2] model is the absence of a hyperglycemic state despite low insulin levels. Intriguingly, Pax6[Leca2] mice not only display lower blood glucose levels throughout the day but also have normal glucose disposal. Upon further examination, we discovered that these mice have alterations in the circadian variation of clock and metabolic genes in the liver. The SCN in hypothalamus is considered to be the central pacemaker for circadian rhythm by synchronizing central and peripheral clocks[54]. Although Pax6[Leca2] mice lack rhythmic changes in clock gene expressions in the hypothalamus, it is unlikely that the SCN plays a role in exerting the observed changes in liver due to the following reasons. First, PAX6 is not expressed in the SCN but rather in the zona incerta region of the hypothalamus[55]. Therefore, any modifications associated with circadian oscillations in the SCN are indirect consequences of the Leca2 mutation, indicating upstream defects as a likely cause. Second, recent studies have investigated diurnal rhythms of the liver transcriptome regulated by light-dark cycles independent of the SCN[56,57]. Remarkably, mice lacking Bmal1 specifically in the SCN retain some rhythmicity in the liver if kept under normal light-dark conditions, showing that endogenous clock entrainment can occur directly due to light[58]. Again, lack of rhythmic changes in liver gene expressions in Pax6[Leca2] mice hint at lack of light-mediated entrainment of the peripheral clock. Additionally, we observed high variability in parameters measured by indirect calorimetry, and in the liver and hypothalamic gene expression patterns for Pax6[Leca2] samples. Hence, this evident loss of synchrony between individual mice occurs likely due to loss of light perception, which is attributable to retinal disorganization of ipRGCs and a lack of optic nerve.

Loss of Bmal1 specifically in hepatocytes reduces hepatic glucose output and increases glucose tolerance in mice[59]. Therefore, loss of circadian rhythm can potentially lead to dampening of liver function, as observed for the Pax6[Leca2] mice. Moreover, loss of Bmal1 and Clock may result in increased insulin sensitivity, possibly due to disruptions of core-clock components in the liver[29,59,60]. Nevertheless, Pax6[Leca2] mice do not inherently lack any of the clock genes and the aforementioned circadian gene as well as related mouse models[29,60,61] do not have intrinsic defects with the light-sensing system. In this regard, del Rio-Martin 2019 et al.[24] recently showed that mice having functional and structural defects in the retina and retinohypothalamic tract showed changes in circadian rhythm and energy metabolism similar to that seen in the Pax6[Leca2] model, including normal glucose tolerance. However, additional studies are required to clearly dissect effects of the Leca2 mutation on the eye and metabolism. Based on these results, we highlight the pleiotropy of PAX6, where the Leca2 mutation leads to loss of circadian rhythm, which in turn seems to nullify any effect that β cell dysfunction may impose upon the physiology of these mice.

## Methods

**Animals.** All mice were kept in a specific-pathogen-free environment in compliance with the Federation of European Laboratory Animal Science Associations (FELASA) protocols. Mice were housed with a 12-h light/dark cycle (6 a.m. to 6 p.m./6 p.m. to 6 a.m.) with free access to food and water. All experimental procedures were performed in accordance with German and European Union guidelines. Pax6[Leca2] mice[19] were provided by Prof. Magdalena Götz and maintained on a C3HeB/FeJ background. All experiments were performed with male homozygous Pax6 mutant mice denoted as Pax6[Leca2] and wild-type littermates as WT.

**Body composition and metabolic studies.** Food was removed at 8 am, 6 h prior to an oral gavage of either 2 g/kg of glucose (20% solution, Braun) or an

intraperitoneal (ip) injection of 0.75 U/kg of human insulin (Lilly). Blood glucose was measured using Akku-Check (Roche) from tail blood before (time point 0) and 15, 30, 60, 90, and/or 120 min after an oral or ip administration and collected in heparinized tubes (Sarstedt). From another animal cohort, blood was collected via tail at four different time points with a 6-hour interval at ZT0 (06:00), ZT6 (12:00), ZT12 (18:00) and ZT18 (00:00) in EDTA-containing tubes (Sarstedt). Ad libitum body weight and blood glucose levels were measured between 11:00–12:00. Body composition analysis was carried out using noninvasive nuclear magnetic resonance to measure fat and lean mass (MiniSpec LF50, Bruker). Indirect calorimetry including activity monitoring, food and water uptake recording was measured by using the PhenoMaster indirect calorimetry system (TSE Systems).

**Micro-computed tomography.** Contrast-enhanced micro-CT data were acquired on a laboratory micro-CT system Bruker Skyscan 1172 (Bruker). A solution of iodine potassium iodide (Lugol solution, Sigma–Aldrich) was used as the contrast agent to increase the differential attenuation of X-rays among the soft tissues[62]. The specimen was fixed in a phosphate-buffered formal saline solution, polymerized formaldehyde dissolved as a 4% solution in phosphate-buffered saline, and then placed in 50% Lugol solution, made by diluting 100% Lugol with deionized H20, for 14 days. This contrast media was replaced every other day. The stained sculls were embedded in paraffin wax in small plastic tubes and placed in the scanner as described in ref. [62]. All scans were acquired with the following parameters: 89 kVp source voltage, 112 μA source current, 1200 ms exposure time, 0.4° increment, 0.5 mm aluminum filter, 8 frame averages, isotropic voxel resolution of 8 μm³. The projection data were reconstructed with the dedicated manufacturer software (InstaRecon). Image visualization and volumetric analysis of the pineal gland were carried out using 3D Dataviewer and the software package 3D Slicer[63].

**Magnetic resonance imaging.** Magnetic resonance imaging (MRI) was performed on a preclinical high-field 9.4 T system Bruker BioSpec 94/20 USR (Bruker) using ParaVision 6.01 imaging software. Volumetric data was acquired using the manufacturer's 4-channel RX array surface coil in combination with a quadrature TX volume resonator. A three-dimensional (3D) gradient-echo sequence (FLASH) was applied, with a protocol as follows: flip angle: 30°, echo time TE: 2.7 ms, repetition time TR: 50 ms, FOV: $15 \times 15 \times 3.75$ mm³, acquisition matrix: $96 \times 96 \times 32$, pixel size: $156 \times 156 \times 117$ μm³, averages: 1.

**Clinical chemistry.** Final blood samples were collected from mice in deep anesthesia (Isothesia, Henry Schein) from vena cava using EDTA-containing S-Monovette® (Sarstedt) and centrifuged in a 1.5 mL tube (Eppendorf) at 10 °C and $7.69 \times g$ in a Biofuge centrifuge (Heraeus) to obtain plasma, which was snap frozen in liquid nitrogen. Several parameters such as nonesterified free fatty acids (NEFA), glycerol, triglycerides (TG), cholesterol, high density lipoproteins (HDL), low density lipoproteins (LDL), lipase, and lactate were analyzed using a Beckman–Coulter AU480 autoanalyzer (Beckman–Coulter) according to a procedure described elsewhere[64].

**Hyperinsulinemic-euglycemic clamp.** The clamp study was performed following a protocol published recently[65]. In brief, mice were anesthetized and cannulated by inserting a permanent catheter into the jugular vein. After recovery for 7 days, food access was restricted 4 h before a 2-h primed-continuous [3–³H] glucose infusion (1.85 kBq/min). Subsequently, hyperinsulinemic-euglycemic clamp were started with a [3–³H] glucose infusion (3.7 kBq/min) containing insulin (8 pmol/kg/min; Humulin R, Lilly) and a variable amount of 20% glucose to maintain euglycemia. Steady state was achieved at minutes 90–120. To estimate insulin-stimulated glucose uptake in individual tissue, 2-deoxy-d-[1-¹⁴C] glucose (370 kBq) were injected intravenously at minutes 120. Blood samples were collected into heparinized tubes at minutes 90, 100, 110, 120, 122, 125, 130, and 140. Urine was collected at the end for glucose measurement as previously described[65]. Mice were euthanized with a lethal dose of ketamine/xylazine and tissues were snap frozen for glucose uptake determination.

**In vitro studies.** Pancreatic islets were isolated as previously described[66] and cultured overnight in RPMI medium (Lonza) supplemented with 11.1 mM glucose (Sigma–Aldrich), 10% v/v FBS (Gibco) and 1% v/v penicillin/streptomycin (Invitrogen). To carry out static in vitro stimulation of insulin secretion, isolated islets were incubated in modified Krebs–Ringer Buffer containing 5 mM HEPES and 24 mM $NaHCO_3$ and 1.5 mM glucose (Sigma–Aldrich) for 2 h at 37 °C. Next, 10 similar sized islets per experiment and mouse were stimulated with low (2.8 mM) and high (12 mM) glucose concentrations as well as 100 nM exendin-4 (Sigma–Aldrich) in combination with high glucose for 2 hours each. To measure hormonal content in the islets, 500 μL of acid-ethanol was added to a separate batch of 10 similar sized and unstimulated islets.

**ELISAs.** Insulin and glucagon in samples obtained from in vitro studies and plasma were measured using mouse ELISA (Mercodia); corticosterone and β-hydroxybutyrate was measured using corticosterone and beta-HB ELISA kit

(Abcam), respectively, and melatonin was measured using Melatonin ELISA Kit (Aviva Systems), all according to manufacturer's instructions.

**Tissue extraction and immunohistochemistry**. Excised pancreases were fixed in 4% PFA, serially incubated in 9, 15, and 30% sucrose solutions and embedded in optimum cutting temperature solution (Thermo Scientific). 10 μm sections were obtained with a Leica CM1850 Cryostat (Leica Microsystems) and 3–4 sections, >300 μm apart, were placed on SuperFrost® Plus slide (Menzel–Gläser) and stored at −20 °C. Euthanized wild-type mice had their eyes enucleated and immersed in 4% PFA for overnight fixation. Socket tissue was collected and prepared from Pax6$^{Leca2}$ mice by excising the orbits with their content after incision cuts performed around the closed lids through the orbits bone. The tissue collected was fixed in 4% PFA overnight and the next day bones were removed from the material. Next, globes and orbits tissue were rinsed in 70% ethanol and embedded in paraffin. Sections 5-μm thick were cut. To perform immunostaining, the tissue sections were permeabilized with 0.1% Tween20/PBS, blocked with 5% BSA and subsequently incubated with primary antibodies overnight at 4 °C and appropriate secondary antibodies at RT for 90 min and mounted in Vectashield® Mounting Medium (Vector Laboratories). All other fluorescent images were obtained using a Leica TCS SP5 laser-scanning confocal microscope and composite images were constructed and analyzed using ImageJ software.

To determine the α and β cell volume, pancreases were fixed in 4% (w/v) neutrally buffered formalin and standardized cut into four pieces before embedded in paraffin in order to achieve a cut surface as large as possible. Islet analysis was performed on two sections of 3 μm, which were 300 μm apart. After a co-staining for insulin and for glucagon nuclei were labelled with Hoechst33342 (Thermo Fischer). The stained tissue sections were scanned with an AxioScan.Z1 digital slide scanner (Zeiss, Jena, Germany) equipped with a ×20 magnification objective. Quantification of insulin or glucagon expression cells were performed by using image analysis software Definiens Developer XD2 (Definiens AG, Germany). α and β cell volume [mg] was calculated by multiplying the detected relative insulin-positive and glucagon-positive cell area by total pancreatic weight, respectively. Used antibodies are provided in Supplementary table 2 and complete images including cropped portions are provided in Supplementary Fig. 7a.

**RNA isolation and expression profiling**. Tissue from liver was homogenized in trizol (Invitrogen) using Heidolph DIAX 900 (Sigma–Aldrich) and total RNA was isolated using the RNeasy Mini kit (Qiagen). Total RNA from isolated islets was extracted using the RNeasy Plus Micro kit (Qiagen). The Agilent 2100 Bioanalyzer in combination with the Agilent RNA 6000 Pico Kit was used to assess RNA quality. Only high-quality RNA (RIN > 7) was used for further analyses. qRT-PCR was performed for relative quantification of genes in cDNA samples using the fluorescent cyanine dye SYBR Green I included in the LightCycler® 480 DNA SYBR Green I Master (Roche) according to the manufacturer's instruction. Primer pairs used are provided in Supplementary Data 2. The results were determined as described elsewhere[67] and relative gene expressions for islets were normalized to housekeeping genes Rpl13a and Fbxw2, for liver to Atp5b and Hmbs, and for hypothalamus to Rpl13a and Tuba1a1. Total RNA (15 ng) was amplified using the Ovation PicoSL WTA System V2 in combination with the Encore Biotin Module (Nugen). Amplified cDNA was hybridized on Affymetrix Mouse Gene 2.0 ST arrays containing about 35,000 probe sets. Staining and scanning (GeneChip Scanner 3000 7 G) were done according to the Affymetrix expression protocol including minor modifications as suggested in the Encore Biotion protocol. Statistical analyses were performed by utilizing the statistical programming environment R[68] implemented in CARMAweb[69]. Transcriptome Analysis Console (4.0.1.36, Thermofisher Scientific) was used for quality control and to obtain annotated normalized RMA gene-level data using standard settings. Genewise testing for differential expression was done employing the limma t-test and Benjamini–Hochberg multiple testing correction (FDR < 10%) and filters for fold-change >1.5× and average expression in at least one group >16 were applied.

**Protein extraction and western blots**. Protein from handpicked isolated islets or liver tissue was extracted by suspending them in ice-cold RIPA Lysis and Extraction Buffer (Thermofisher Scientific) supplemented with 1× cOmplete® Mini Protease Inhibitor Cocktail (Roche) and PhosSTOP™ (Roche). Protein concentrations were obtained using Pierce BCA Protein Assay Kit (Thermofisher Scientific) according to manufacturer's instructions. 10–30 μg of protein sample was loaded onto a 10% SDS-polyacrylamide gel (Bio-Rad). Protein was then transferred to a nitrocellulose membrane (Thermofisher Scientific). The membrane was blocked with Odyssey® Blocking Buffer (TBS) (LI-COR), thereafter incubated overnight at 4 °C in primary antibody and in secondary antibody for 45 min at RT in the same buffer. For fluorescent detection of proteins, Odyssey Infrared Imaging System and Odyssey® software (LI-COR) was utilized. Densitometric quantification of western blot image was performed using Image Studio Lite version 5.2 (LI-COR) and expressed as relative fluorescence intensity. Used antibodies are provided in Supplementary table 2 and complete images including cropped portions are provided in Supplementary Fig. 7b, c.

**Statistics and reproducibility**. Statistical analysis was achieved using GraphPad Prism 8 software and applied using two-tailed unpaired Student's or Welch's t-test, one-way, two-way or Welch's ANOVA with Bonferroni's or Dunnett's post-hoc test to compare two or more groups. D'Agostino & Pearson test was used to determine normal distribution of data acquired by indirect calorimetry. Body composition data and metabolic rate was compared applying linear regression modelling with body mass as a co-variate. CircWave analysis (Roelof A. HUT, Department of Chronobiology, University of Groningen, Netherlands) was used to generate regression curves to determine the probability of circadian rhythmicity of temporal gene expression pattern. p values thus attained are indicated in appropriate graphs. A value of $p < 0.05$ was considered significant. All results are described either as means ± Standard error of mean (s.e.m.) or ±standard deviation (s.d.) as stated in figure legends. Sample number designated by "n" represents number of individual mice. All experiments could be reproduced with similar results.

**Reporting summary**. Further information on research design is available in the Nature Research Reporting Summary linked to this article.

## Data availability

Raw transcriptomics dataset described in the current study data can be found in the GEO database with accession number GSE128504. All data generated or analyzed to support the current study are provided in the published article and online supplementary data (Supplementary Data 1–3) or can be obtained from the corresponding author upon reasonable request.

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

## Acknowledgements
We thank Prof. Magdalena Götz (Helmholtz Zentrum München, Institute of Stem Cell Research) for providing the Pax6$^{Leca2}$ mouse model, sharing ideas, and fruitful discussions. We thank M. Kraiger and M. Sandholzer (Helmholtz Zentrum München, German Mouse Clinic) for performing the MR scans and µCT analysis. We also thank Ann-Elisabeth Engelniederhammer, Michael Schulz, Andreas Mayer, and Sandra Hoffman (Helmholtz Zentrum München, Institute of Experimental Genetics and German Mouse Clinic, Neuherberg, Germany) for providing excellent technical assistance. This work was supported by the German Center for Diabetes Research (DZD) and from the Helmholtz Alliance 'Aging and Metabolic Programming, AMPro' (J.B.).

## Author contributions
N.F.C. designed the study, carried out research, collected data, and wrote the manuscript. M.W., M.R., D.G., M.I., A.L.A., O.V.A., A.F., and B.R. performed experiments and analyzed the data. J.R., G.P., E.W., and J.B. supervised and coordinated experiments. J.R., P.H., R.T., G.P., and M.H.d.A. conceived and designed the study, contributed to the interpretation of results, and critically reviewed the manuscript. G.P. and M.H.d.A. are the guarantors of this work and, as such, had full access to all the data in the study and take responsibility for the integrity of the data and the accuracy of the data analysis. All authors approved the final version of the manuscript.

## Funding

## Competing interests

The authors declare no competing interests.
