## [Peer Review File · Communications Biology]

Reviewers' comments:

Reviewer #1 (Remarks to the Author):

A relationship between PAX6, circadian rhythm, and β cell function is well established, and this group previously showed that PAX6 alleles in mouse leading to the small-eye phenotype. In this current manuscript the authors explore the relationship between PAX6 and β -cell function without affecting glucose tolerance in mouse. In an interesting series of experiments they found despite beta cell dysfunction and decreased insulin secretion are associated with reduced hepatic glucose production by altered circadian variation in circadian clock genes and metabolism gene and then adding on further *leca2* mutant revealing pleiotropic effect of PAX6. The phenotype characterization of the β cells from PAX6 mutation is due to RED subdomain of PAX6. The mechanistic studies showing decreased hepatic glucose production by lack of rhythm change in clock and metabolism gene are novel and interesting although it is not clear the molecular mechanism. Nevertheless, the results are informative and clearly show an unexpectedly important role for the PAX6 protein in the regulation of hepatic glucose production and the islet transcriptome in a PAX6*leca2* mice. Few points should be carefully described and interpreted prior to publication.

Minor issues:

It is surprising that this article focuses on a circadian rhythm gene but does not address time of day relative to fasting/feeding cycles, in its phenotyping. Since the main phenotype, hepatic glucose, is largely affected by fasting and feeding, processes that have circadian rhythms and are strongly regulated by fasting and feeding, it is critical to know what conditions were used and ideally, to test both circumstances.

It seems that only male mice were studied. Is there a valid rationale for this choice? It is well known that sex hormones influence rodent metabolism.

Figure 3 and Figure 4 mice were male or female?

What was the light cycle?

Reviewer #2 (Remarks to the Author):

The authors report on a point mutation (R128C) in the RED subdomain of PAX6, which has been described in a human patient, to present a comprehensive study of a homozygous Pax6 mutation in the context of adult mammalian metabolism and circadian rhythm.

The point mutation has been reported to have homozygous viability in mice, perhaps the only one so far as homozygosity of PAX6 mutations normally are DEADLY

It is a very interesting and extensive study in many ways, and I have enjoyed reading several of the sections. However, it is unclear to me how the authors can put so much weight on what goes on in the eyes of these mice. Homozygous Pax6*Leca2* mice not only lack eyes at the embryonic stage, their eyelids are closed throughout life. Thus, the eye structures of these mice cannot be expected to be anything like normal. Normal eyes develop from early embryonic stages and this development continues throughout adolescence of mammals and it comes as no surprise that the retina is not normal in these mice. Retinal development continues after birth and epigenetic factors are thought to play a role. With underdeveloped retinas it is also expected that ipRGCs have not developed properly. This may not be the main problem; with closed eyelids there is also a limitation with regards to amount and spectral composition of the light reaching the retina. Furthermore, in a normal retina the ipRGCs make out less than 5% of the RGCs – to what degree are the authors certain that the number of RGCs are close to normal? How do the cornea and crystalline lens look in these mice? The degree of transparency and whether the lens is separated from the cornea or not would also play a role if any light passes through the closed eyelids. Any of

these structures and their properties will also act as spectral filters affecting the wavelength composition of the light reaching the retina. I am therefore not convinced that "the mutation leads to loss of circadian rhythm in Pax6Leca2 mice likely due to incorrect distribution and organization of ipRGCs." To what degree are between mice differences related to differences in anterior segment development?

Details with regards to figures:

Figure 1. The variation in Pax6Leca2 mice is large and none of the correlations are significant. There are no error bars, but it states n=4.

Figure 4d. Data shown only for one WT and one Pax6Leca2 – but it states n = 3?

Figure 6. The variation in Pax6Leca2 mice is large and none of the correlations are significant.

Reviewer #3 (Remarks to the Author):

Pax6 is one of the most important developmental transcription factors required for the development of the eye and various portions of the brain. The authors describe the phenotype in adult Pax6 mutations carrying a homozygous mutation in a particular domain RED, that does not cause prenatal or postnatal death of the animals, atypical feature of most Pax6 mutants. The manuscript is well written although the discussion could have been a little more contextualised with the phenotypes of various circadian rhythm gene mutations to explain the link between circadian rhythm loss due to lack of visual entrainment and the metabolic phenotype. The results are interesting, but the authors need to explain 2 primary questions to complete the story:

1. How does the RED domain mutation alter the protein structure that leads to the altered phenotype? Is it changing the binding partners? What is its status inside the cells compared to WT? Where is it located? A ChIP seq would have been a better experiment than simply mRNA profiling in both the hypothalamus and the beta cells to establish whether it is directly controlling the circadian rhythm genes and the metabolic genes. The authors must bear in mind that mutations in various genes like Clock, Bmal1, Cry1 etc cause a variety of metabolic dysfunctions based on the mutation types, so how does the Leca2 phenotype match with those?
2. The link between circadian rhythm and altered metabolism appears more of an association rather than a mechanistic one at this point. In order to prove that the metabolic phenotype is due to the circadian rhythm anomaly, a conditional mutant that does not express in the eye can be tested. Something that has been shown for Clock gene mutations. Or they can complement WT Pax6 in the islet cells and show reversal of phenotype. Alternatively, the authors can consider rescue experiments with drugs or other downstream genes or siRNA that can suppress the effect of the mutant Pax6 to see if the insulin-glucose levels and ketone body formation may be reversed. The authors demonstrate the phenotype variations such as ketone body formation, altered energy production pathway genes without further validation that can help the reader arrive at a singular cohesive mechanism of the phenotype described.

Additional comments and concerns.

- How does the phenotype in the human RED domain mutation patients compare with the mouse in terms of metabolism and circadian rhythm entrainment? Is the human phenotype recessive and do they display DSPD or insomnia typical of disturbances in circadian rhythm entrainment?
- What is the phenotype in the Pax6-Leca2 heterozygous mice? Showing data from the heterozygous littermates might also be useful. Are there any motor function deficiencies in the heterozygous and homozygous mutant mice?
- In earlier studies comparing the different Pax6 mutations and their associated transcriptional profiles, have the circadian genes been found to be altered?
- What is the status of Pax4 which has been shown to repress Pax6 expression in pancreas? Since it has been reported that Pax6 expression does not vary with circadian cycle, but Pax 4 expression does vary diurnally, the authors should establish what happens with Pax4 in this animal model.

- Along with an insulin secretagogue GLP-1 agonist, a rescue experiment that induces proinsulin mRNA expression itself should be tested.
- Do levels of IGF isoforms have a role in the reduced insulin expression? Also, in the absence of GLUT2, the status of the other GLUT receptors needs to be evaluated. Do these mice have higher GLUT4 expression in muscles and adipose tissues? How is the muscle metabolism affected? Does that counterbalance the hepatic phenotype since the mutant animal does not show hyperglycemia despite lower insulin levels?

Point-by-point response to the reviewers:

First of all, the authors would like to thank all the reviewers for the time and effort they have taken to comment on our manuscript by Chhabra et al. and acknowledging their interest in our study. We have carefully considered each point of criticism and suggestion to provide a point-by-point response below. Overall, we believe that the paper has now considerably improved following this revision. Please note that updated or newly added figures are provided at the end of the document.

Reviewers' comments

A relationship between PAX6, circadian rhythm, and β cell function is well established, and this group previously showed that PAX6 alleles in mouse leading to the small-eye phenotype. In this current manuscript the authors explore the relationship between PAX6 and β -cell function without affecting glucose tolerance in mouse. In an interesting series of experiments, they found despite beta cell dysfunction and decreased insulin secretion are associated with reduced hepatic glucose production by altered circadian variation in circadian clock genes and metabolism gene and then adding on further *leca2* mutant revealing pleiotropic effect of PAX6. The phenotype characterization of the β cells from PAX6 mutation is due to RED subdomain of PAX6. The mechanistic studies showing decreased hepatic glucose production by lack of rhythm change in clock and metabolism gene are novel and interesting although it is not clear the molecular mechanism. Nevertheless, the results are informative and clearly show an unexpectedly important role for the PAX6 protein in the regulation of hepatic glucose production and the islet transcriptome in a PAX6*leca2* mice. Few points should be carefully described and interpreted prior to publication.

Reviewer #1:

Reviewer comments and questions	Authors' response
1. It is surprising that this article focuses on a circadian rhythm gene but does not address time of day relative to fasting/feeding cycles, in its phenotyping. Since the main phenotype, hepatic glucose, is largely affected by fasting and feeding, processes that have circadian rhythms and are strongly regulated by fasting and feeding, it is critical to know what conditions were used and ideally, to test both circumstances.	We apologize for not having presented the experimental conditions in a way that is better to understand. We have specified the time and lighting conditions when we first mentioned the term "zeitgeber" in the results section in line 121. In addition, all necessary information is given in the methods section in line 401 and in the graphs.
2. It seems that only male mice were studied. Is there a valid rationale for this choice? It is well known that sex hormones influence rodent metabolism.	The reviewer has correctly noted that only male mice were used. The authors are aware of the sexual dimorphisms in animal models that translate into differences in phenotypes (Karp et. al., 2017 Nat Comm ; Dearden et. al., 2018 Mol Metab ; Gannon M., 2018 Mol Metab). However, the same studies also point out that

	female mice tend to be somewhat resistant to metabolic phenotypes or show weaker phenotypes than males, thereby making gene-phenotype interactions difficult to study (Tiano and Mauvais-Jarvis., 2012, Nat Rev Endocrinol). Moreover, comparable studies (e.g. Del Rio-Martin et al., ref. 24; Husse et al., ref. 54; Lamia et al., ref. 57; Marcheva et al., ref. 58) are mostly carried out with male mice. Therefore, for better comparison, only male mice were used.
3. Figure 3 and Figure 4 mice were male or female?	Only male mice were used for all experiments in this study.
4. What was the light cycle?	Unless otherwise stated, all mice were kept under specific pathogen-free conditions with a 12-hour light/dark cycle (6 a.m. to 6 p.m./6 p.m. to 6 a.m.) with free access to food and water. We have added this information in the methods section in line 388.

Reviewer #2:

Reviewer comments and questions	Authors' response
1. The authors report on a point mutation (R128C) in the RED subdomain of PAX6, which has been described in a human patient, to present a comprehensive study of a homozygous Pax6 mutation in the context of adult mammalian metabolism and circadian rhythm. The point mutation has been reported to have homozygous viability in mice, perhaps the only one so far as homozygosity of PAX6 mutations normally are DEADLY.	Indeed, Pax6 mouse models studied thus far show homozygous lethality in the postnatal stage. In the literature, one exception exists. Homozygous viability was observed for the Pax6 ^{14Neu} model, which harbors an amino acid substitution in the homeodomain. However, homozygous Pax6 ^{14Neu} mice do not show any observable metabolic or pancreatic phenotype (Dames 2010, ref. 8). This has been mentioned in the discussion from line 286.
2. It is a very interesting and extensive study in many ways, and I have enjoyed reading several of the sections. However, it is unclear to me how the authors can put so much weight on what goes on in the eyes of these mice. Homozygous Pax6 ^{Leca2} mice not only lack eyes at the embryonic stage, their eyelids are closed throughout life. Thus, the eye structures of these mice cannot be expected to be anything like normal. Normal eyes develop from early embryonic stages and this development continues throughout adolescence of	We thank the reviewer for acknowledging the extensive work done on this model. Pax6 ^{Leca2} mice are on a C3HeB/FeJ background. As discussed in lines 296-301, these mice have an inherent Pde6b gene mutation (https://www.jax.org/strain/000658) that causes loss of photoreception, resulting in lack of image formation. Hence, RGCs in this context are neither relevant in mutants nor in WT mice. This has been discussed in lines 296-301. The rest of the eye is not an “eye” but rather an accumulation of cells and tissues. To provide further evidence in support of our

mammals and it comes as no surprise that the retina is not normal in these mice. Retinal development continues after birth and epigenetic factors are thought to play a role. With underdeveloped retinas it is also expected that ipRGCs have not developed properly. This may not be the main problem; with closed eyelids there is also a limitation with regards to amount and spectral composition of the light reaching the retina. Furthermore, in a normal retina the ipRGCs make out less than 5% of the RGCs – to what degree are the authors certain that the number of RGCs are close to normal? How do the cornea and crystalline lens look in these mice? The degree of transparency and whether the lens is separated from the cornea or not would also play a role if any light passes through the closed eyelids. Any of these structures and their properties will also act as spectral filters affecting the wavelength composition of the light reaching the retina. I am therefore not convinced that “the mutation leads to loss of circadian rhythm in Pax6Leca2mice likely due to incorrect distribution and organization of ipRGCs.” To what degree are between mice differences related to differences in anterior segment development?	claim, we have added MRI images and have shown a lack of correct eye development in the mutant model. Please see supplementary fig. 1a for a comparative view. Additionally, we investigated the brain and eye morphology to determine any changes in the connection between the hypothalamus and the retina. We found a lack of the optic chiasm and optic nerve in mutant mice. This would essentially not allow for any relay of information from the eye to the hypothalamus regarding the status of time of day. We have added these images to the supplementary results (Supplementary fig. 1b) and made relevant textual changes in results and discussion in lines 100, 114, 281 and 311 as well as in methods section in line 422.
3. Figure 1. The variation in Pax6Leca2 mice is large and none of the correlations are significant. There are no error bars, but it states n=4.	The reviewer has correctly noted that error bars are not displayed and the variation is high within the mutant group. Precisely to overcome this issue, we have used “CircWave analysis” [Roelof A. HUT, Department of Chronobiology, University of Groningen, 507 Netherlands] that provides us with a p value, which indicates a significant change in the temporal regulation of the mRNA content within an experimental group. As shown in Figure 1b and Figure 6a,b, change of mRNA level within the WT group over time is significant, while that in the mutant group is not. This is due to a large variation in the mutant group as seen in the graphs where all individual data points are displayed. This suggests absence of circadian rhythmicity in mutants. p values are provided in the graphs

	and an explanation of the analysis and the reference is provided in methods in line 549.
4. Figure 4d. Data shown only for one WT and one Pax6Leca2 – but it states n = 3?	Thank you for pointing it out. We have included images from individual WT and mutant mice in the Figure 4d.
5. Figure 6. The variation in Pax6Leca2 mice is large and none of the correlations are significant.	Please refer to the explanation for comment 3. An explanation of the analysis and reference is provided in the methods section in line 549.

Reviewer #3:

Pax6 is one of the most important developmental transcription factors required for the development of the eye and various portions of the brain. The authors describe the phenotype in adult Pax6 mutations carrying a homozygous mutation in a particular domain RED, that does not cause prenatal or postnatal death of the animals, atypical feature of most Pax6 mutants. The manuscript is well written although the discussion could have been a little more contextualised with the phenotypes of various circadian rhythm gene mutations to explain the link between circadian rhythm loss due to lack of visual entrainment and the metabolic phenotype. The results are interesting, but the authors need to explain 2 primary questions to complete the story:

Reviewer comments and questions	Authors' response
1. How does the RED domain mutation alter the protein structure that leads to the altered phenotype? Is it changing the binding partners? What is its status inside the cells compared to WT? Where is it located? A ChIp seq would have been a better experiment than simply mRNA profiling in both the hypothalamus and the beta cells to establish whether it is directly controlling the circadian rhythm genes and the metabolic genes. The authors must bear in mind that mutations in various genes like Clock, Bmal1, Cry1 etc cause a variety of metabolic dysfunctions based on the mutation types, so how does the Leca2 phenotype match with those?	In the study on the Leca2 model by Walcher 2013 (ref. 20), the authors reported that an in silico analysis suggested disruption of DNA binding by the RED subdomain, located within the sixth helix of paired domain, rather than change in protein structure. Moreover, the same authors and others (Yamaguchi 1997, ref. 47) have shown that the R128C mutation reduces PAX6 activity driven via the 5aCON site, therefore affecting DNA binding. With regards to this, we saw a downregulation of the preproglucagon gene (Gcg) in isolated islets of Pax6^{Leca2} mice (Supplementary figure 2b), transcription of which is driven via this site (Singer 2019, ref. 48). This example is discussed from line 329. PAX6 was found to be located inside the nucleus. For a comparative view, we have carried out PAX6 staining on pancreatic sections and confocal images are provided in Supplementary fig. 4b. Relevant textual changes have been made in line 191.

The reviewer has raised an important point regarding the metabolic dysfunctions associated with mutations in important clock genes. While the Pax6^{Leca2} model shows loss of circadian rhythm similar to *Clock*, *Bmal1*, *Cry1* and related models (Marcheva 2010, ref. 58, Rudic et. al., 2004, *Plos Biol*, Kalsbeek 2014, ref. 29), two major differences persist between them. First, these models do not have intrinsic problems with the light-sensing system. Second, Pax6^{Leca2} mice do not inherently lack any of the clock genes. Instead, the data shows that their temporal regulation is out of sync with light. Therefore, we have taken caution in a direct comparison between the two types of models since it might lead to speculation. Moreover, mRNA expression profiling of isolated islets of global clock gene model, Clock^{Δ19} (Marcheva 2010, ref. 58) have little overlap (five genes namely, *Bcd1*, *Pfkip*, *Hook1*, *igh-VJ558* and *Il15ra*) with that of the Pax6^{Leca2} model. Therefore, a more reliable comparison would be that with a mouse model with a distorted function of melanopsin positive cells (ipRGCs) and lack of hypothalamic innervation. In this regard, del Rio-Martin 2019 (ref. 24) recently showed that mice having functional and structural defects in the retina and retinohypothalamic tract showed changes in circadian rhythm and energy metabolism similar to that seen in the Pax6^{Leca2} model.

Thank you for the suggestion. ChIP-Seq will certainly help us to determine direct effects of the Leca2 mutation on the beta cell function. Unfortunately, beta cells cannot be reliably purified from murine islets without a transgenic reporter mouse line, for which we currently neither have the legal nor the technical capacity under the global circumstances. Regarding the hypothalamus, we have performed ChIP-Seq analysis on the whole tissue and have found a small number of enriched genes reflecting changes in binding targets of PAX6. However, PAX6 is expressed only in the *zona incerta* region of the hypothalamus (Chen 2017, ref. 53) and the

	ChIP-Seq data may not truly reflect changes in PAX6 binding and may lead to dubious conclusions. Moreover, any molecular changes in circadian rhythm, for which the master clock lies in the suprachiasmatic nucleus, must be indirect since PAX6 is not expressed there. Finally, in light of the recent finding that Pax6^{Leca2} mice lack the optic nerve and optic chiasm (Supplementary fig. 1b), the data further supports the claim that the hypothalamus may only have an indirect role to play. (Please see lines 343-363 for a discussion on the matter). Respectfully, at this time with the current data set, we do not consider changes in the hypothalamus as direct effects of the Leca2 mutation and as relevant to the main message of the paper.
2. The link between circadian rhythm and altered metabolism appears more of an association rather than a mechanistic one at this point. In order to prove that the metabolic phenotype is due to the circadian rhythm anomaly, a conditional mutant that does not express in the eye can be tested. Something that has been shown for Clock gene mutations. Or they can complement WT Pax6 in the islet cells and show reversal of phenotype. Alternatively, the authors can consider rescue experiments with drugs or other downstream genes or siRNA that can suppress the effect of the mutant Pax6 to see if the insulin-glucose levels and ketone body formation may be reversed. The authors demonstrate the phenotype variations such as ketone body formation, altered energy production pathway genes without further validation that can help the reader arrive at a singular cohesive mechanism of the phenotype described.	Indeed, a conditional knockout model could help us reveal a direct mechanistic link between the circadian rhythm and the metabolic phenotype. Complementing WT PAX6 to isolated islets could also help confirm a direct effect of the Leca2 mutation on the islet function. With regards to this, the glucose stimulated insulin secretion assay (Figure 3g) and transcriptome analysis (Figure 4a-c), was performed on isolated islets. Considering that these experiments were performed in vitro after an overnight incubation period (methods lines 454-463), one can posit that the results indicate, at least in part, a more direct effect of the mutation in the islets rather than effects originating from the whole-body phenotype. We thank the reviewer for these suggestions however, considering the production of the mouse model and subsequent phenotyping will take over 6 months as well as establishing in vitro techniques under the current minimum working capacity is unfortunately not possible.
 • How does the phenotype in the human RED domain mutation patients compare with the mouse in terms of metabolism and circadian rhythm entrainment? Is the human phenotype recessive and do they display DSPD or insomnia typical of 	As stated in a study by Azuma 1996 (ref. 18), R128C results in a dominant phenotype, where several members of the same family, displayed foveal hypoplasia. The study further reports that “all patients had normal growth, intelligence, physical examination and karyotypes”. However, no further detail is provided.

disturbances in circadian rhythm entrainment?	Several clinical variations at the R128 position of PAX6 namely; R128H and R128L have been reported in the NCBI database in association with Aniridia. Another variant R128S has been reported without any clinical significance. Bredrup et. al., 2008 Arch Ophthalmol reported R128P mutation causing peripheral corneal opacities, correctopia, iris hypoplasia, early cataract formation, highly variable axial lengths, and foveal hypoplasia in afflicted patients. Although no further detail on the patients was provided, there seems to be a propensity for mutations at the R128 site in humans. Some association of insomnia in a cohort of Korean subjects has been reported with the PAX6 gene (Ban et. al., 2011, Plos One). Pineal gland hypoplasia has been reported in some patients with PAX6 mutations (Mitchell 2003 ref. 27, Hanish 2016 ref. 28 and Berntsson et. al., 2019 J Sleep Res). Therefore, we performed micro-computed tomography (μCT) and carried out volumetric analysis on the pineal gland in WT and Pax6^{Leca2} mice. We found a strong decrease in pineal gland volume indicating hypoplasia. The representative data has been added to supplementary figure 2f. Relevant textual changes have been made from line 153 and 308 as well as methods in line 407.
 • What is the phenotype in the Pax6-Leca2 heterozygous mice? Showing data from the heterozygous littermates might also be useful. Are there any motor function deficiencies in the heterozygous and homozygous mutant mice? 	Gross morphological differences were not found between the heterozygous and WT mice. The study by Walcher 2013 (ref. 20) on the Leca2 model also only made use of homozygous mice. Additionally, a pilot study was carried out with heterozygous littermates and the data indicated no significant difference in glucose metabolism. Comparative values of body weight and blood glucose levels have been added to the supplementary fig. 3a,b. Relevant textual changes have been made in lines 161.
 • In earlier studies comparing the different Pax6 mutations and their associated transcriptional profiles, have the circadian genes been found to be altered? 	In the available literature regarding Pax6 mouse models (carrying point mutations) that studied this gene in the context of islet function or metabolism, no report exists for an extensive transcriptome profiling. However, two reports

	that independently investigated a beta cell specific Pax6 knockout model carried out transcriptome analysis. Mitchell 2017 (ref. 17) carried out RNA-Seq on isolated islets but did not report any circadian genes. Swisa 2018 (ref. 16) reported downregulation of Nr1d1 gene in islet transcriptome. However, the data set was acquired at a single time point in the murine Min-6 cell line. Other studies such as Walcher 2013 (ref. 20) carried out ChIP-Seq and microarray analysis on cortical tissue of the Leca2 model, but did not report any circadian genes in their data set. Xie et. al., 2013 Plos One, reported ChIP-chip data on the beta-TC3 cell line and did not report any circadian gene targets of PAX6.
 • What is the status of Pax4 which has been shown to repress Pax6 expression in pancreas? Since it has been reported that Pax6 expression does not vary with circadian cycle, but Pax 4 expression does vary diurnally, the authors should establish what happens with Pax4 in this animal model. 	The reviewer is correct in pointing out that Pax4 cycles in the retinal photoreceptors (Rath et. al., J Neurochem 2008). However, Perelis et. al., 2015 Science, reported that both Pax4 and Pax6 do not cycle in the islets. Moreover, as mentioned in the methods section in line 454, the standard procedure of islet isolation does not allow for studying temporal changes in islet cells. We only extract information on the islet transcriptome at a single time point after an overnight incubation period, which provides the islets a recovery period after the harsh collagenase treatment. Of note, Pax4 expression was not found to be altered in the islet transcriptome of the Leca2 model (Supplementary table 1).
 • Along with an insulin secretagogue GLP-1 agonist, a rescue experiment that induces proinsulin mRNA expression itself should be tested. 	Although not a rescue experiment as recommended by the reviewer, we attempted to stimulate insulin secretion in isolated islets using the GLP-1 agonist exendin-4. As described in Figure 3g, exendin-4 increased insulin secretion to some extent in islets of mutant mice in vitro but did not rescue the phenotype.
 • Do levels of IGF isoforms have a role in the reduced insulin expression? Also, in the absence of GLUT2, the status of the other GLUT receptors needs to be evaluated. Do these mice have higher GLUT4 expression in muscles and adipose tissues? How is the 	IGF-I is mainly secreted by the liver and has insulin-like action in lowering blood glucose and improving insulin sensitivity (Clemmons et. al., 2003 JCI). However, in the clamp experiment, we show that insulin itself dramatically lowers hepatic glucose output

muscle metabolism affected? Does that counterbalance the hepatic phenotype since the mutant animal does not show hyperglycemia despite lower insulin levels?

(Figure 5f). Furthermore, we did not check the expression of GLUT4 in muscle or adipose tissue since the clamp experiment clearly showed no change in glucose uptake under the influence of insulin between mutant and WT mice (Figure 5e). Additionally, no change was found in the lipolytic substrates (glycerol and NEFAs, supplementary table 2), suggesting normal metabolic status of white adipose tissue. The data suggests that reduced hepatic glucose production is the main reason behind normal glucose tolerance in the *Leca2* mutants without any change in muscle or fat insulin sensitivity. A similar phenotype has been reported for mice lacking the circadian gene *Bmal1* in liver (Lamia 2008, ref. 57) or *Gcgr* gene (Sorensen et. al., 2006 *Diabetes*; Gelling et. al., 2003 *PNAS*) that show reduced hepatic glucose output results in either normal or improved glucose tolerance. This phenotype in *Gcgr* knockout mice has been reported even in the absence of functional beta cells (Lee et. al., 2011 *Diabetes*).

Figure 4 | *Leca2* mutation leads to changes in islet transcriptome. (a) Heat map displaying differentially expressed genes (DEGs) in isolated islets. Genes were filtered for fold-change >1.5x and FDR <10%. n=4. (b) A comparative analysis of DEGs in *Leca2* mutants and known PAX6 targets. (c) Relative islet mRNA expression of *Pax6*, *Ins2*, *Pcx*, *Ffar1*, *Mafa* and *Ucn3*. n=4 (WT *Ucn3* n=3, *Leca2* *Pcx* n=3), ns – non-significant, * $p < 0.05$, ** $p < 0.01$, *** $p < 0.001$ Welch's and Student's *t* test. (d) Representative immunofluorescence images for GLUT2 in pancreatic islets. n=3. Scale bars, 50 μ m. (e) Representative immunofluorescence images displaying insulin and glucagon positive cells in islets and quantifications of α and β mass, and their ratio; n=5, >80 islets per mouse were analyzed. Scale bars, 100 μ m. * $p < 0.05$, ** $p < 0.01$ Welch's *t*-test and Student's *t* test. 10-12-week-old mice were used for all experiments. Error bars display \pm s.d.

a**WT****Leca2****b**
Supplementary figure 1: Lack of optic nerve and optic chiasm in homozygous Pax6^{Leca2} mice. (a) Representative MRI images displaying eye and eye structures in WT and mutant mice (white asterix). (b) Ventral aspect of the brain displaying presence of optic nerve and optic chiasm in the WT and lack thereof in mutants (white dotted box). 10-12-week old male mice were used for this study.

Supplementary figure 3: Changes in food intake and RER in Pax6^{Leca2} mice. 6-hour fasted measurements of (a) body weight and (b) blood glucose levels (WT n=12, Het=12, Hom=13). ** $p < 0.01$, *** $p < 0.001$ one-way or Welch's ANOVA followed by Bonferroni's or Dunnett's *post hoc* test. 12-week-old male mice were used for this study. Error bars display \pm s.e.m.

Supplementary figure 4: Leca2 mutation does not affect islet architecture. (a) Representative western blot images and quantification of PAX6 expression. WT n=6, Leca2 n=7. ***p<0.001 Student's *t* test. (b) Representative images of nuclear expression of PAX6 in pancreatic islets. (c) Relative islet mRNA expression of genes encoding hormones in 10-week-old male mice. n=4, *p<0.05 Student's *t* test. (d) Representative immunofluorescence images of cells positive for insulin and proliferation marker Ki-67 (arrows), and respective quantifications thereof in (e). ~60 islets per genotype were analyzed, n=3, Scale bars, 20 μ m. (f) Plasma glucagon levels after a 6-hour fasted period. WT n=10, Leca2 n=8. Error bars display \pm s.d. No significant change was found in (e) using Student's *t* test.

REVIEWERS' COMMENTS:

Reviewer #1 (Remarks to the Author):

I am satisfied with the revisions and the increased clarity regarding the caveats of interpretation.

Reviewer #2 (Remarks to the Author):

I thank the authors for thorough revision.

Reviewer #3 (Remarks to the Author):

The answers and additional data as well as references have answered many of the queries raised though not all. In particular, I feel a rescue experiment with something other than extendin-4 would have strengthened the conclusions. I understand that not all questions may be answered during these extraordinary times. However, in the final version of the manuscript, I would like to see some of these critical points they have discussed in the rebuttal and expand the statements as limitations of the study. Key things in the discussion to be enhanced:

1. The comparison of the other circadian rhythm genes and metabolism with their model where an association is shown.
2. The discussion about the mutant localisation and function with respect to the known phenotype in the patients.

The authors should also consider tempering their conclusions a bit, particularly, lines 322-326, 377-379.

Point-by-point response to the reviewers:

The authors would like to again thank all the reviewers for the time and effort they have taken to comment on our manuscript by Chhabra et al. and acknowledging their interest in our study. We have added the final changes to the manuscript as recommended by the reviewer 3. We feel the manuscript is ready in its final form for publication.

Reviewer 1

Reviewer comments and questions	Authors' response
I am satisfied with the revisions and the increased clarity regarding the caveats of interpretation.	The authors thank the reviewer for their time and effort

Reviewer comments and questions	Authors' response
I thank the authors for thorough revision.	The authors thank the reviewer for their time and effort

Reviewer comments and questions	Authors' response
The answers and additional data as well as references have answered many of the queries raised though not all. In particular, I feel a rescue experiment with something other than extendin-4 would have strengthened the conclusions. I understand that not all questions may be answered during these extraordinary times. However, in the final version of the manuscript, I would like to see some of these critical points they have discussed in the rebuttal and expand the statements as limitations of the study. Key things in the discussion to be enhanced: 1. The comparison of the other circadian rhythm genes and metabolism with their model where an association is shown.2. The discussion about the mutant localisation and function with respect to the known phenotype in the patients. The authors should also consider tempering their conclusions a bit, particularly, lines 322-326, 377-379.	The authors thank the reviewer for their time and effort. 1. Necessary changes have been made in the discussion to highlight the differences between mouse models of clock genes and the Pax6^{Lcca2} model in the context of light-sensing mechanism and circadian rhythm.2. We have added a note on different PAX6 mutations in human patients at the R128 site. For the aforementioned textual changes, we mention the need for further investigations and experiments. 3. The conclusions and claims in lines 322-326 and 377-379 have been changed to avoid any speculations.